# A Systemic Review of the Integral Role of TRPM2 in Ischemic Stroke: From Upstream Risk Factors to Ultimate Neuronal Death

**DOI:** 10.3390/cells11030491

**Published:** 2022-01-31

**Authors:** Pengyu Zong, Qiaoshan Lin, Jianlin Feng, Lixia Yue

**Affiliations:** 1Department of Cell Biology, Calhoun Cardiology Center, University of Connecticut School of Medicine (UConnHealth), Farmington, CT 06030, USA; zong@uchc.edu (P.Z.); feng@uchc.edu (J.F.); 2Department of Ecology and Evolutionary Biology, University of Connecticut, Storrs, CT 06269, USA; qiaoshan.lin@uconn.edu

**Keywords:** ischemic stroke, TRPM2, oxidative stress, Ca^2+^ signaling

## Abstract

Ischemic stroke causes a heavy health burden worldwide, with over 10 million new cases every year. Despite the high prevalence and mortality rate of ischemic stroke, the underlying molecular mechanisms for the common etiological factors of ischemic stroke and ischemic stroke itself remain unclear, which results in insufficient preventive strategies and ineffective treatments for this devastating disease. In this review, we demonstrate that transient receptor potential cation channel, subfamily M, member 2 (TRPM2), a non-selective ion channel activated by oxidative stress, is actively involved in all the important steps in the etiology and pathology of ischemic stroke. TRPM2 could be a promising target in screening more effective prophylactic strategies and therapeutic medications for ischemic stroke.

## 1. Introduction

### 1.1. Stroke

Globally, stroke is a common cause of death and mortality only secondary to myocardial infarction, and is the third-most-common cause of disability [1]. There are two different types of strokes, ischemic stroke and hemorrhagic stroke. Ischemic stroke is much more common than hemorrhagic stroke, accounting for over 80% of all the cases [1,2]. The pathological feature of ischemic stroke is occlusion of supplying arteries and subsequent tissue necrosis [3]. The occlusion of brain arteries can result from two causes, emboli formed outside the brain (extracranial) or thrombus formed in situ (intracranial) [2]. Emboli from atherosclerotic plaque on carotid arteries or cardiac thrombus from atria during atrial fibrillation is the most common cause [3,4]. In situ thrombosis resulting from local atherosclerosis is less common but still an important cause of ischemic stroke [3]. Small-vessel diseases caused by hypertension and diabetes can cause small subcortical infarcts [2,4].

Following occlusion, all the cells “downstream” to the occlusion site are subjected to ischemic conditions, including endothelial cells, glial cells, and neurons. Endothelial hyperpermeability leads to vasogenic edema and immune-cell infiltration, which significantly aggregate brain damage and increase the mortality of patients [5,6,7]. The role of glial cells in ischemic stroke is controversial [8,9]. In some studies, activation of reactive astrocytes and microglia increases brain injury after ischemic insult [10,11]. In other cases, astrocytes and microglia were also proved to be neuro-protective and critical in tissue repair after ischemic stroke [12,13,14]. However, the underlying molecular mechanisms regulating endothelial permeability and glial activation after ischemic stroke remains obscure. 

The ultimate pathological change in ischemic stroke is neuronal death, which is initiated in neurons themselves and aggravated by glial cell activation and immuno–inflammatory responses [15]. Neurons are the most ischemia-susceptible cells in human body—irreversible cellular injury and cell death happen within minutes following complete oxygen deletion [15]. For decades, glutamate excitotoxicity mediated by N-methyl-D-aspartate (NMDA) receptor has been thought to be the culprit of neuron death during ischemic stroke [16,17]. However, the failure of targeting NMDA receptors in the treatment of ischemic stroke is shifting researchers’ attention to exploring other molecular mechanisms contributing to neuron death [16], among which TRPM2 emerges as a pivotal molecule in aggregating neuron death (Figure 1).

### 1.2. TRPM2

Transient receptor potential melastatin 2 (TRPM2) is a Ca^2+^-permeable and non-selective ion channel [18,19,20] belonging to the TRP channel family [21,22,23]. This 172 kDa membrane protein is located in an ~90 kb area on chromosome 21q22.3 in humans [24]. Compared with other members of the TRPM family, TRPM2 shares the characteristic N-terminal homology regions (MHR1–MHR4) and C-terminal coiled-coil domains (CTD), but is featured by its unique C-terminal NUDT9-H domain [25]. The NUDT9-H domain is homologous to NUDT9, a highly conserved adenosine diphosphate ribose (ADPR) pyrophosphatase [25,26]. Therefore, TRPM2 can be referred as a chanzyme. The enzyme function of NUDT9-H varies depending on the species [27,28]. Although invertebrate NUDT9-H is an active ADPRase, vertebrate NUDT9-H does not have the ADPRase activity. However, the NUDT9-H domain is found to be critical for the surface expression of TRPM2 and the gating of TRPM2 by ADPR [29,30]. Besides the cell membrane, TRPM2 is also suggested to be localized in the membrane of lysosomes [31,32].

Upon activation, TRPM2 currents display a characteristic linear I–V relationship, with a single channel conductance about 75–78 pS [33]. TRPM2 is gated by ADPR and Ca^2+^, and can be inhibited by N-(p-amylcinnamoyl) anthranilic acid (ACA) [34] and 2-aminoethoxydiphenyl borate (2-APB) [34]. Recent atomic structural analysis has revealed a Ca^2+^ binding site [35,36,37] which plays a major role in channel gating for several TRPM channels including TRPM2, TRPM4, and TRPM8 [38]. Activation of TRPM2 also requires ADPR binding to the channels in the presence of Ca^2+^. It was well known that ADPR binds to the C-terminal NUDT9-H domain of TRPM2 [39]. Interestingly, recent TRPM2 structures reveal that in zebrafish (*Danio rerio*) TRPM2 (*dr*TRPM2), besides the NUDT9-H domain, there is also another ADPR binding site at the N terminal MHR1/2 domain [40]. However, this newly discovered binding site in *dr*TRPM2 is not involved in the ADPR gating in human TRPM2 [40], although the antagonist 8-Br-cADPR binds to the MHR1/2 domain. Like many other TRP channels, TRPM2 is a temperature sensor [41]. TRPM2 is activated by heat, and the temperature threshold of TRPM2 is 47.2 ± 0.2 °C, but this threshold can be significantly reduced to a physiologically reachable level (36.3 ± 0.6 °C) by oxidative stress [42,43]. Moreover, oxidative stress also indirectly activates TRPM2 by increasing the production of ADPR and Ca^2+^ [44]. Therefore, TRPM2 is regarded as a cellular sensor for oxidative stress [45]. The local temperature in the affected tissue during inflammation usually increases [46], and oxidative-stress-mediated Ca^2+^ signaling is critical for the elicitation of inflammatory responses in immune cells [47]. The combined sensing of heat and oxidative stress confers TRPM2 with a crucial function in regulating inflammatory responses (Figure 2 and Figure 3).

TRPM2 is ubiquitously expressed in almost all tissues and cell types [48], and TRPM2-mediated Ca^2+^ signaling is involved in various important cellular functions, including cytokine/hormone secretion [31,49], cytoskeletal rearrangement [32], cell migration [50], regulation of reactive oxygen species (ROS) production [51], autophagy [52], inflammasome activation [49], and cell death [53]. Therefore, TRPM2 is closely related to many human diseases, such as myocardial infarction [54], ischemic stroke [55,56,57], Alzheimer’s disease [58,59], cardiomyopathy [60], atrial fibrillation [61], hypertension [62], atherosclerosis [63], inflammatory lung injury [51,64], diabetes [65], ischemic kidney disease [66], and many cancers [67]. Besides ischemic stroke itself, almost all the above diseases are the upstream etiological factors for ischemic stroke [68], highlighting the critical and comprehensive role of TRPM2 in the development and progression of this devastating disease. 

## 2. *Trpm2* in Diseases Increasing Risk for Ischemic Stroke

The TOAST classification denotes the causes of ischemic stroke into five subtypes: large-artery atherosclerosis, cardio-embolism, small-vessel occlusion, other determined etiology, and undetermined etiology, which is widely accepted by clinicians [69]. Among all the etiological factors of ischemic stroke, the most common ones are atrial fibrillation, hypertension, atherosclerosis, diabetes, and thrombosis [1,2]. Well control of these diseases is important for the prevention of ischemic stroke. However, the molecular mechanisms involved in the development of these diseases are still not completely understood. Here, we give a concise review of the role of TRPM2 in the development and/or progression of these etiological factors of ischemic stroke (Figure 1).

### 2.1. Atrial Fibrillation

Atrial fibrillation (AF) is a common disease in the elderly, and atrium-derived thrombus caused by AF is one of the most common causes of ischemic stroke [70]. Atrial remodeling is the cellular mechanism promoting the development and maintenance of AF [71], in which atrial fibroblasts play a predominant role by increasing the production of extracellular matrix proteins, thereby causing atrial fibrosis [72]. 

The principal risk factor for AF is aging [73]. Aging is usually closely related to chronic systemic inflammation, which is referred as inflammaging [74]. Chronic inflammatory response is an critical driving force in the development of atrial remodeling by enhancing the fibrotic activity of atrial fibroblasts [75]. TRPM2 is an important regulator of inflammation. Previously, TRPM2 was found to be associated with age-associated inflammatory responses in the brain, and deletion of TRPM2 protected mice against age-associated cognitive function caused by inflammation [76]. Moreover, loss of glutathione, a physiological antioxidant, during neuron senescence facilitates TRPM2 activation [77]. Aging-related chronic inflammation is also critical in the development of many cardiovascular diseases [74], and previously we found that the expression of TRPM2 was significantly increased in atrial fibroblasts isolated from patients with AF compared with that from non-AF patients [78]. Therefore, TRPM2 might also play an important role in the aging-related chronic inflammation in the atria, thereby promoting the development of atrial remodeling and AF.

Oxidative stress promotes AF development by impairing the contractility of atrial myocytes [79,80] and accelerating atrial fibrosis [81]. In a recent paper we showed that TRPM2 activation markedly and rapidly promoted the production of ROS in mitochondria of macrophages [63]. TRPM2 itself is a cellular sensor for oxidative stress [45]. Therefore, the increased production of ROS will in turn further promote the activation of TRPM2, thereby forming a feed-forward vicious cycle [63]. Moreover, Ca^2+^ signaling is critical for the activation of fibroblasts during atrial fibrosis [82,83], and TRPM2-mediated Ca^2+^ influx under oxidative stress has been shown to be required for many cellular functions [31,49,50,52]. Considering the high expression of TRPM2 in atrial tissue after AF [78], there is a high possibility that TRPM2 also contributes to the progression of AF by magnifying the oxidative stress response and Ca^2+^ signaling in atrial myocytes and fibroblasts.

Intracellular Ca^2+^ is not only critical for regulating the mechanical and electrical activity of healthy atrial muscle, but it also plays an important role in the triggering of AF [84]. AF can be triggered by afterdepolarizations [84]. There are two types of afterdepolarizations, early afterdepolarizations (EAD) and delayed afterdepolarizations (DAD). Both EAD and DAD can cause abnormal electrical activities in atrial muscle, and Ca^2+^ overload during EAD in atrial muscle was shown to trigger AF [85]. The molecular mechanisms of EAD remain mysterious. Some studies suggest that EAD might result from the Ca^2+^ influx from L-type Ca^2+^ channels or spontaneous Ca^2+^ release from the endoplasmic reticulum (ER) [86]. Considering the important role and active involvement of aging and oxidative stress in the development of AF and TRPM2 in these two conditions, TRPM2-mediated Ca^2+^ influx or Ca^2+^ release from lysosome might also contribute to the Ca^2+^ overload during EAD and the triggering of AF.

### 2.2. TRPM2 in Hypertension

Hypertension, perhaps the most prevalent disease in humans, is an independent risk factor for ischemic stroke [87]. Traditionally, hypertension was thought to cause stroke mainly in the elderly and young males [88,89]. Recently, a small increase in blood pressure even as mild as 10 mm Hg was shown to be associated with a 38% increased risk of stroke in females [90]. The characteristic pathological change of hypertension is arteriosclerosis, which increases the peripheral resistance and blood pressure [91]. The progressive hardening of arteriole walls is called arterial remodeling [92], in which endothelial dysfunction and smooth-muscle proliferation play a central role [91].

Oxidative stress has long been known to promote the development and progression of hypertension by inducing endothelial dysfunction [93], smooth muscle hypertrophy [94], and vascular remodeling [95], whereas antioxidants were shown to have a protective effect against hypertension by preventing the vascular dysfunction [96]. However, the underlying molecular mechanisms remain unknown [97]. In a recent study, the Ca^2+^ dysregulation and hyperactivity of vascular smooth muscle cells during hypertension was found to be dependent on TRPM2-mediated Ca^2+^ influx, which is activated by angiotensin-II-mediated increase of ROS production [62]. Similarly, endothelial dysfunction mediated by ROS-activated TRPM2 was also found to accelerate the development of Alzheimer’s disease [58] and aggregate inflammatory lung injury [64]. As a cellular sensor for oxidative stress, TRPM2 might mediate the detrimental effects of ROS on endothelial cells and smooth muscle cells in the development of hypertension. 

Intracellular Ca^2+^ is a critical regulator of endothelial function [98] and smooth muscle contractility [99]. Ca^2+^ signaling regulates the expression of endothelial nitric oxide synthase [98,100], whereas blockade of Ca^2+^ channels enhanced the production of nitric oxide, a potent vasodilator, in endothelial cells [101,102]. Moreover, increase of intracellular Ca^2+^ in smooth muscle cells enhanced muscle tone and peripheral resistance [103]. In platelets isolated from hypertensive patients, cellular Ca^2+^ concentration was much higher than that from patients with normal blood pressure [104]. There are two sources of cytosolic free Ca^2+^—one is the Ca^2+^ influx from the extracellular environment, and another is the Ca^2+^ release from intracellular organelles, such as the endoplasmic reticulum and lysosomes. The multiple roles of TRPM2-mediated Ca^2+^ influx have been well documented in many cell types including endothelial cells [64]. However, there are a limited number of studies reporting the function of TRPM2 in lysosomes. The lysosome is the most important organelle for autophagy and TRPM2 has long been shown to be associated with autophagy [105]. Lysosomal TRPM2-mediated Ca^2+^ release was shown to be responsible for the pancreatic β-cell death under oxidative stress [31]. Recently TRPM2-mediated Ca^2+^ release from lysosomes was found to promote autophagic degradation in vascular smooth muscle cells, thereby causing cell death [106], and knockout of TRPM2 attenuated hypertension in spontaneously hypertensive rats by reconstituting autophagy in endothelial cells and vascular smooth muscle cells [107]. In summary, TRPM2-mediated Ca^2+^ signaling aggregates the dysfunction of endothelial cells and vascular smooth muscle cells in the development and progression of hypertension.

### 2.3. TRPM2 in Atherosclerosis

Atherosclerosis is a dangerous risk factor for ischemic stroke. Atherosclerotic plaque on the aortic arch [108] and carotid artery [109] significantly increases the risk of ischemic stroke. Moreover, break of atherosclerotic plaque in cerebral arteries directly leads to the formation of in situ thrombus [110], which is usually more difficult to evaluate and predict due to the small plaque size and deep position in the skull [110]. The central pathological feature of atherosclerosis is foam cell formation [111]. Uptake of too much cholesterol transforms infiltrated macrophages into highly inflammatory foam cells, which are the culprit in the development and progression of atherosclerotic plaque [112]. Foam-cell formation includes two critical processes—one is macrophage infiltration, the another is phagocytosis of cholesterol included in oxidized low-density-lipoprotein (oxLDL) [112].

The first step of macrophage infiltration during atherosclerosis is macrophage chemotaxis toward the lesion site, which is caused by chemokines. Previously, TRPM2 was found to be critical for the chemotaxis of neutrophils by formyl-methionyl-leucyl-phenylalanine (fMLP) [113]. fMLP is also a well-established chemokine for macrophages, suggesting the potential role of TRPM2 in regulating macrophage chemotaxis. Indeed, in a recent study, we found that the in vitro macrophage migration induced by macrophage chemotaxis protein 1 (MCP1) was inhibited by deleting TRPM2 or inhibiting the activation of TRPM2 [63]. MCP1 secreted by endothelial cells in response to subendothelial deposition of oxLDL is one of the initial driving forces of macrophage infiltration during atherosclerosis [112]. Similar to our findings, hydrogen peroxide (H_2_O_2_) was shown to attract neutrophil migration both in vivo and in intro, which was also abolished by TRPM2 knockout or TRPM2 inhibition [114]. H_2_O_2_ is an important molecular signal generated during inflammation [115]. H_2_O_2_ gradient produced by wounded tissue is required for the rapid leukocyte recruitment after injury [116]. Like the chemotactic effect on neutrophils, H_2_O_2_ might also be an important chemokine for macrophages, and the recruitment of macrophages mediated by H_2_O_2_ might depend on TRPM2 activation. 

The second step of macrophage infiltration into the vessel wall is attachment and penetration of the endothelium, which is called extravasation [112]. Leukocyte extravasation is mediated by several surface-adhesion molecules. TRPM2 was found to be required for the transendothelial migration of neutrophils induced by endotoxin, in which TRPM2-mediated Ca^2+^ influx promotes the phosphorylation of VE-cadherin and degradation of tight junctions between endothelial cells [50]. Similarly, specific deletion of TRPM2 in immune cells markedly decreased immune-cell invasion into the brain after ischemic stroke [55]. Moreover, in mice fed with high-fat diet, *Trpm2* deletion caused reduced macrophage infiltration and attenuated inflammation in adipose tissue compared with wild type mice [117]. In our study, we also found that TRPM2-mediated Ca^2+^ influx is also needed for the in vitro transendothelial migration of macrophages induced by MCP1, and *Trpm2* deletion reduces the macrophage burden in atherosclerotic plaques in vivo [63]. These studies highlight the crucial role of TRPM2 in promoting macrophage infiltration during atherogenesis.

After infiltrating into the vessel wall, macrophages engulf oxLDL and become highly proinflammatory foam cells [112]. Foam cells promote the development and progression of atherosclerosis by secreting pro-inflammatory cytokines, chemokines, and tissue-degrading enzymes, which cause profound inflammatory responses and lead to lesion expansion [111]. Activation of the NFκB signaling pathway is required for the activation of macrophages, and TRPM2-mediated Ca^2+^ signaling has been shown to be indispensable for NFκB signaling activation in macrophages during inflammation, suggesting the potential tole of TRPM2 in transforming macrophages into foam cells [118]. CD36 is the most important receptor for oxLDL uptake in macrophages, and activation of the CD36 downstream signaling pathways is required for macrophage activation and subsequent foam-cell formation [119,120,121,122,123,124]. Previously TRPM2 was shown to mediate the activation of macrophages during infection [125,126] and inflammation [49,127] or when temperature increases [42]. We found that TRPM2 is required for CD36 activation and oxLDL uptake in macrophages, and activation of CD36 by oxLDL further promotes the activation of TRPM2, thereby forming a feed-forward viscous cycle [63]. NLRP3 inflammasome activation by engulfed cholesterol is required for macrophage activation during atherosclerosis [128]. We also found that NLRP3 inflammasome activation by oxLDL is dependent on TRPM2-mediated Ca^2+^ signaling [63]. All these studies suggest TRPM2 is likely to play a crucial role in the transformation of infiltrated native macrophages into pro-inflammatory foam cells.

### 2.4. TRPM2 in Diabetes

Hyperglycemia causes a series of pathological changes in the vessel walls, including endothelial dysfunction, basal membrane thickening, interstitial fibrosis, and vessel stiffness, which markedly increase the risk of developing ischemic stroke [129]. Moreover, the mortality of ischemic stroke is higher and clinical outcomes are poorer in patients with diabetes [130]. Well-controlled hyperglycemia significantly decreases the risk of ischemic stroke, decreases mortality, and improves clinical outcomes [131]. The prominent pathological feature of diabetes is insufficient insulin secretion (type 1 diabetes mellitus, T1DM) or unresponsiveness of peripheral tissues to insulin (type 2 diabetes mellitus, T2DM). Many studies have demonstrated that TRPM2 is implicated in the development of both T1DM and T2DM.

T1DM is featured by gradual loss of pancreatic β cells by chronic inflammation in pancreatic islets, thereby resulting in lack of insulin secretion [132]. Ca^2+^ influx is required for insulin secretion [133], which is further amplified by intracellular Ca^2+^ release [134]. TRPM2 has a high expression level in pancreatic β cells and activation of TRPM2-mediated Ca^2+^ influx by ADPR is required for insulin secretion, which can be further enhanced when temperature increases [135]. Knockout of TRPM2 leads to decreased serum insulin levels, increased glucose levels in plasma, and higher insulin sensitivity of peripheral tissues [65,117]. However, excessive Ca^2+^ influx also leads to cell death under pathological conditions. Both TRPM2-mediated Ca^2+^ influx and lysosomal Ca^2+^ release were found to be involved in β-cell death caused by H_2_O_2_ [31,136]. Autoimmune inflammatory responses play a central role in the destruction of β cell during T1DM [132]. During chronic inflammation, infiltrated self-reactive T cells and macrophages can produce substantial amount of H_2_O_2,_ and oxidative stress is also an important mechanism for β-cell dysfunction during T1DM [132,137,138,139]. Moreover, infiltrated immune cells also secrete cytokines such as tumor necrosis factor-α (TNF-α) [140], and TRPM2-mediated Ca^2+^ influx has been shown to mediated the cytotoxic effect of TNF-α [20,31]. Therefore, TRPM2 is a key molecule in promoting β-cell death in the development of T1DM.

The distinguishing feature of T2DM is insulin resistance [141]. Peripheral tissues, especially adipose tissues, lose their ability to respond to insulin, which results in decreased uptake of glucose in the blood [142]. Macrophage-mediated chronic inflammation is a major mechanism for the reduced insulin resistance in adipose tissue [143,144,145,146,147], and reduced macrophage load in adipose tissue is associated with better blood-glucose control in human patients [148]. In mice fed with a high-fat diet, *Trpm2* deletion alleviated inflammation and macrophage infiltration and improved insulin resistance in adipose tissue compared with wild-type mice [117]. As mentioned earlier, TRPM2-mediated Ca^2+^ influx is required for macrophage infiltration and activation during inflammation [42,49,127] or under different pathological conditions [50,63,125,126]. Therefore, TRPM2 activation might also be required for macrophage infiltration into adipose tissue and the subsequent activation. Targeting TRPM2 can be a promising strategy to inhibit adipose tissue inflammation and improve insulin sensitivity during T2DM. 

### 2.5. TRPM2 in Thrombosis

Extracranial thrombus (from the left atrium) and intracranial in situ thrombosis are important causes of ischemic stroke [1,2]. Platelet aggregation plays a central role in thrombus formation [147,149,150], during which increase of intracellular Ca^2+^ is required [151,152]. Oxidative stress promotes the recruitment and activation of platelets, but the underlying molecular mechanisms remain unclear [153]. TRPM2 is expressed in megakaryocytes and platelets, but the function of TRPM2 in megakaryocytes and platelets remains mysterious [154,155]. The important roles of TRPM2-mediated Ca^2+^ signaling in many cellular functions have been well documented, and TRPM2 might also be involved in platelet activation during thrombosis. 

Thrombospondin-1 (TSP1) is a potent platelet activator by inhibiting intracellular nitric oxide signaling, and TSP1-CD36 ligation increases thrombus stability [153,156]. We found that TSP1-mediated macrophage activation depends on TRPM2 activation under oxidative stress [63]. Recently, we also found that TRPM2-mediated Ca^2+^ influx is required for TSP1 mediated endothelial dysfunction during ischemic stroke (unpublished). Platelets themselves have an abundant storage of TSP1 in intracellular α-granules, and during platelet activation TSP1 is rapidly released [156]. Moreover, after ischemic stroke, the synthesis and release of TSP1 in the brain was dramatically increased even within hours [157]. Therefore, it is very likely that TRPM2 can promote thrombus formation, and the platelet aggregation induced by TSP1 also needs the involvement of TRPM2.

## 3. Mechanisms by Which TRPM2 Increases Brain Injury during Ischemic Stroke

Compared with all other tissues, the brain has the highest expression of TRPM2 under physiological conditions [48]. Moreover, the expression of TRPM2 after ischemic stroke was significantly increased [57], highlighting the critical role of TPRM2 in promoting brain damage. The central pathological feature of ischemic stroke is neuronal-cell death, and all therapies for ischemic stroke are aimed at preventing or decreasing the loss of neurons [2]. The failure of directly targeting neurons in treating ischemic stroke shifts researcher’s attention to a relative new concept, the neurovascular unit (NVU) [16,158,159]. The NVU is composed of endothelial cells, pericytes, vascular smooth muscle cells, surrounding glial cells (mostly astrocytes), and related neurons [160]. After ischemic stroke, all the components of the NVU are subjected to hypoxic condition and dysfunction of non-neuron cells significantly aggregates neuron death [2]. Studies have shown that TRPM2 activation in each component of the NVU can aggregate brain damage during ischemic stroke (Figure 4, as summarized in Table 1).

### 3.1. TRPM2 and Cerebral Endothelial Hyperpermeability

Anatomically, cerebral endothelial cells are the core components of the NVU [161]. To fulfil the function of underlying the outer surface of the blood–brain barrier (BBB), cerebral endothelial cells (CECs) have several unique properties compared with peripheral endothelial cells, including densely distributed tight junctions and significantly higher mitochondrial content [162]. After ischemic stroke, degradation of tight junctions between CECs causes plasma extravasation and immune-cell infiltration, and the resulting cerebral edema and inflammatory damage significantly increase the mortality of patients [5,6,7,163]. However, the molecular mechanisms causing endothelial dysfunction during ischemic stroke remain unclear [164].

The roles of TRPM2 in regulating the permeability of peripheral endothelial cells have been extensively studied. In one study, transendothelial migration of neutrophils across lung-derived endothelial cell was shown to be dependent on the activation of TRPM2 in endothelial cells [50]. TRPM2-mediated Ca^2+^ influx dissembled adherens junctions and opened paracellular pathways, thereby allowing for neutrophil extravasation [50]. Another study showed that H_2_O_2_ treatment results in hyperpermeability in lung-derived endothelial cells by increasing ADPR production thereby activating TRPM2 and causing Ca^2+^ overload, and inhibition of TRPM2 produced a potent protective effect against H_2_O_2_-induced endothelial hyperpermeability [64]. Since the mitochondrion is the major organelle producing ADPR under oxidative stress, the higher mitochondria content in CECs could result in more robust ADPR production compared with lung-derived endothelial cells [162]. Thus, TRPM2 is likely to be activated under this condition. Indeed, we also found that oxygen–glucose deprivation (OGD), an in vitro mimic of ischemic stroke, induced substantial ROS production and hyperpermeability in CECs, whereas *Trpm2* deletion or TRPM2 inhibition protected CECs from OGD-induced Ca^2+^ overload, tight-junction degradation, and endothelial hyperpermeability (under revision). Moreover, we also found that endothelial-specific deletion of TRPM2 significantly alleviated plasma extravasation and immune-cell infiltration after middle cerebral artery occlusion (MCAO), in an in vivo mimic of human ischemic stroke in mice (under revision).

### 3.2. TRPM2 and Immune-Cell Invasion

Normally, the densely distributed tight junctions between CECs prevent the migration of circulating immune cells into the brain [6]. During ischemic stroke, however, massive immune-cell infiltration occurs due to the degradation of tight junctions and secretion of chemokines following tissue injury, which causes profound inflammatory responses and aggregated brain damage [5,6]. As discussed earlier, TRPM2 is essential for immune cell migration in various pathological situations [42,49,50,63,117,125,126,127]. Indeed, replacement of bone marrow in wild-type mice with bone marrow from *Trpm2* knockout mice significantly decreased infiltration of neutrophils and macrophages into the brain and reduced infarction size after MCAO, indicating TRPM2 activation in immune cells is required for their penetration across the BBB after ischemic stroke [55]. After entering into the brain, immune cells are activated and secrete cytotoxic cytokines, such as TNF-α and IL-1β, which further promote neuron death [6]. Considering TRPM2-mediated Ca^2+^ signaling is required for immune cell activation and cytokine production [42,49,50,63,117,125,126,127], targeting TRPM2 in immune cells might provide extra benefit in mitigating tissue damage following immune cell extravasation.

### 3.3. TRPM2 in Glial Cells

Microglia and astrocytes are the two most important glial cells involved in tissue inflammation and repair after ischemic stroke [9]. Microglia are the brain-resident close relatives of peripheral macrophages. Under physiological conditions, microglia are responsible for clearing cell debris and shaping synapses during brain development [165]. Under pathological conditions such as ischemic stroke, activated microglia phagocytize dead cells and cause inflammatory responses [166]. When activated, ROS production in microglia is increased [167]. Increased ROS is not only needed for activation of pro-inflammatory genes, but microglia also secrete excessive ROS into surrounding tissues, which causes comprehensive neuron damage [167]. As mentioned earlier, ROS activates TRPM2 by increasing the production of ADPR, and TRPM2-mediated Ca^2+^ signaling is required for the activation of macrophages, the peripheral cousins of microglia. In addition, the expression of TRPM2 in microglia was increased after H_2_O_2_ treatment [168]. Therefore, TRPM2 is likely to contribute to microglia activation after ischemic stroke. Indeed, H_2_O_2_ and ADPR activate TRPM2 in cultured microglia [169], and TRPM2 is required for TNF-α production in microglia [170,171]. Deletion of TRPM2 also significantly alleviated brain damage in a chronic cerebral hypoperfusion mouse model by inhibiting microglial activation [172]. Moreover, CD36 activation in microglia promoted ROS production and aggregated brain injury after ischemic stroke [173]. Recently, we also found that activation of CD36 in macrophages depends on TRPM2-mediated Ca^2+^ influx [63]. Thus, TRPM2 may also be required for CD36 activation in microglia.

Astrocytes are the major housekeeping cells in the brain. Their functions include recycling released neuro-transmitters in synapses, regulating water homeostasis in the brain, secreting neurotrophic factors supporting neuron development, and forming the backbone of the BBB [174]. After ischemic stroke, resting astrocytes are activated by tissue damage and become reactive astrocytes [175]. Proliferation of reactive astrocytes leads to the formation of a glial scar, which is called reactive astrogliosis [138]. Moreover, reactive astrocytes were found to enhance immune response and aggregate brain damage [176]. However, molecular mechanisms regulating astrocyte activation are still not clear. Similarly to activated microglia, ROS production is also increased in reactive astrocytes [177,178,179], and inhibiting TRPM2 suppressed astrocyte activation and reduced the secretion of pro-inflammatory cytokines by astrocytes in response to depletion of glutathione, a ROS neutralizer [171]. The results of above studies strongly suggest the critical role of TRPM2-mediated Ca^2+^ signaling in promoting the detrimental microglial/astrocyte activation and microglial/astrocyte-mediated tissue damage after ischemic stroke.

### 3.4. Neuronal TRPM2 and Neuron Death

Neurons are the most important cells for an intelligent human being, but they are also the most fragile cells when exposed to hypoxia. Neuron death is the hallmark of ischemic stroke. Molecular mechanisms underlying neuron death during ischemic stroke have been extensively studied. Among these mechanisms, two that are important are ROS production and Ca^2+^ overload [1]. Compared with the robust activation of immune cells by ROS and Ca^2+^ influx, even a moderate increase of intracellular ROS and Ca^2+^ can kill neurons [180]. The role of Ca^2+^ influx mediated by oxidative-stress-activated TRPM2 in cell death has been well documented [20,53]. H_2_O_2_ induced Ca^2+^ overload in cultured neurons and caused rapid neuron death, whereas knockdown [181,182], inhibition [182,183], and knockout [56] of TRPM2 produced a potent protective effect against Ca^2+^ overload and neuron death induced by H_2_O_2_ treatment or OGD. IN addition, infarction size was reduced in *Trpm2* knockout mice compared with wild-type mice [184,185,186]. Moreover, TRPM2 was found to be related to delayed neuron death after ischemic stroke, and TRPM2 deletion prevented post-stroke cognitive dysfunction in mice [56]. Recently, we also found that selective deletion of TRPM2 in neurons protects mice from ischemic brain damage by inhibiting ROS production and Ca^2+^ overload [57].

Excitotoxicity caused by NMDA-receptor (NMDAR) activation plays a central role in inducing neuron death during ischemic stroke [1]. Like TRPM2, NMDAR is also a Ca^2+^-permeable channel and Ca^2+^ overload mediated by NMDAR has long been thought to be responsible for neuron death during ischemic stroke [187]. However, due to the complex nature of NMDAR, the contribution of NMDAR to neuron death depends on their composition and location. NMDAR are tetramers consisting of different subunits (GluN1, GluN2a, GluN2b, GluN2c, GluN2d, GluN3a, and GluN3b). There must be two GluN1 subunits in a NMDAR tetramer, while the other two subunits can be any combination of GluN2a-d and/or GluN3a-b [188]. The most abundant and physiologically important subunits in adults are GluN2a and GluN2b [188]. In ischemic stroke, GluN1/GluN2a promotes neuron survival, whereas GluN1/GluN2b causes neuron death [189,190]. In addition, the location of NMDAR is critical in determining the fate of neuron during ischemia. Synaptic NMDAR is crucial in maintaining the normal function of the brain, and is found to protect neuron from ischemic injury [191]. In contrast, NMDAR at extra-synaptic sites mediates long-term depression, and is critical in mediating the glutamate excitotoxicity during ischemic stroke [189]. TRPM2 was found to increase the expression of GluN2b while decreasing the expression of GluN2a, and knockout of TRPM2 reversed this change and protected mice from ischemic stroke [192]. Moreover, in our recent study, we found that TRPM2 was able to physically and functionally associate with GluN2a and GluN2b in NMDAR at extrasynaptic sites, and specific disruption of this TRPM2-NMDAR interaction using a interfering peptide produced a protective effect against ischemic stroke similar to *Trpm2* deletion [57]. The above studies indicate that TRPM2 activation in neurons during ischemic stroke promotes neuron death.

## 4. Other TRPM Channels in Ischemic Stroke

In addition to TRPM2, TRPM4 and TRPM7 are also highly expressed in the brain, and were found to aggravate the brain injury following ischemic stroke. Similar to TRPM2, TRPM4 is activated by intracellular Ca^2+^ [195]. TRPM4 can also be activated by depleting intracellular ATP, suggesting its activity may play an important role in cellular damage when cells undergo energy imbalance during ischemia [195,196]. After MCAO, expression of TRPM4 in the endothelial cells was markedly upregulated, and knockdown of TRPM4 protected mice against ischemic brain injury, suggesting TRPM4 activation may promote BBB degradation after ischemic stroke [197]. Moreover, in a recent study, TRPM4 was found to interact with the NMDA receptors (NMDAR), which enhanced the excitotoxicity and increased neuronal death [198]. Disruption of TRPM4–NMDAR interaction significantly attenuated the MCAO induced brain injury [198]. Compared to TRPM4, the detrimental effects of TRPM7 in ischemic stroke have been extensively studied since 2003 [199]. The unique feature of TRPM7 among all TRPM channels is its high permeability to zinc [199], and zinc toxicity has been well documented as an important cause of neuronal death after ischemic stroke [200]. Knockdown of TRPM7 suppressed the in vivo neuronal death and prevented the post-stroke cognitive dysfunction in mice [201].

## 5. Perspective

Currently, the only two effective therapies for ischemic stroke are intravenous thrombolysis and endovascular thrombectomy. However, both of them have many limitations, including a tight therapeutic window, a low success-rate, and lethal complications [1,2]. For over 30 years, NMDAR-mediated glutamate excitotoxicity was thought to be the major mechanism responsible for neuronal death, but NMDAR antagonists all failed to show any protective effect in clinical trials. In recent years, TRPM2 has been shown to be involved in the pathophysiology of ischemic stroke in all the aspects, including accelerating the development of upstream risk factors outside the brain and aggregating the post-stroke pathological damage inside the brain. Therefore, TRPM2 could be a promising target for screening more effective therapies for preventing ischemic strokes and for treating patients with ischemic stroke. 

There are still two major issues impeding the translation of these exciting findings from basic studies into clinical applications. Firstly, TRPM2 has long been thought to lack of specific inhibitors. However, in recent years, the complete revelation of the high-resolution atomic structure of TRPM2 by cryo-EM has paved the way for developing more specific and effective TRPM2 inhibitors. Secondly, the wide expression of TRPM2 also increases the risk of side effects caused by pharmacological inhibition of TRPM2 during treatment. Luckily, the rapid advancement of nanomedicines has led to the development of more targeted, controlled, and efficient drug delivery systems [202]. Combination of TRPM2-specific inhibitors with highly tissue/cellular specific nano-delivery vehicles could make possible the application TRPM2-inhibition therapies in treating ischemic stroke and its “upstream” diseases.

## Figures and Tables

**Figure 1 cells-11-00491-f001:**
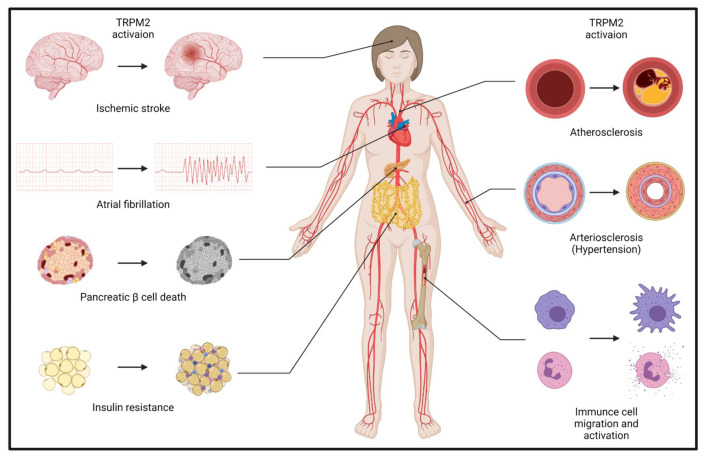
The comprehensive role of TRPM2 in ischemic stroke.

**Figure 2 cells-11-00491-f002:**
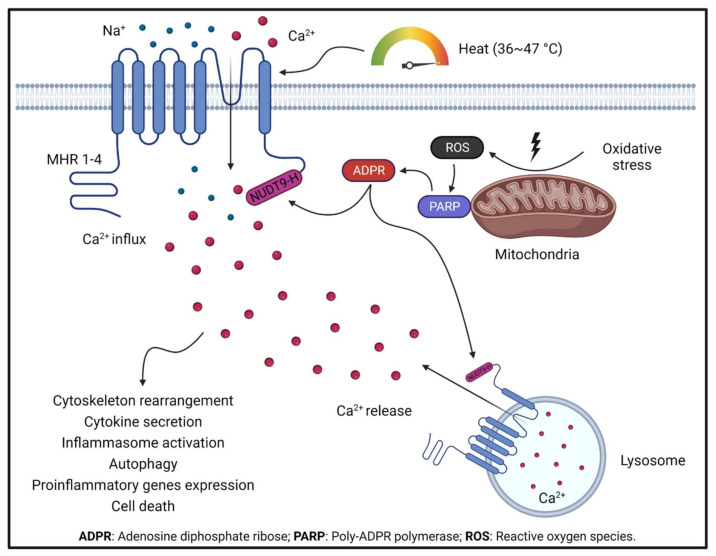
TRPM2 activation by oxidative stress and TRPM2-mediated Ca^2+^ signaling under oxidative stress boosts ROS production in mitochondria. Increased ROS production activates PARP, which produces ADPR, a potent endogenous TRPM2 activator. ADPR activates TRPM2 by binding to the NUDT9-H domain at the C terminus. TRPM2 activation leads to Ca^2+^ influx from the extracellular environment and Ca^2+^ release from lysosomes. TRPM2-mediated Ca^2+^ signaling is critical in regulating a series of cellular functions.

**Figure 3 cells-11-00491-f003:**
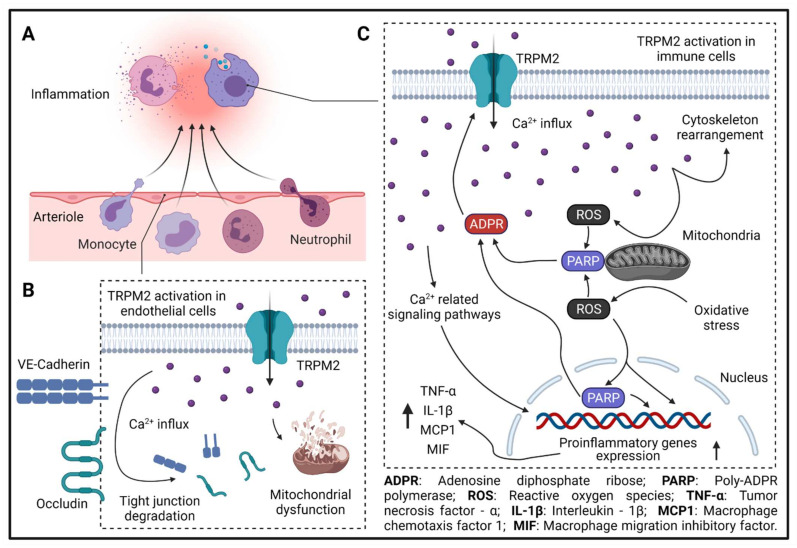
TRPM2 in inflammation. (**A**) Leukocyte extravasation during inflammation. (**B**) TRPM2-mediated Ca^2+^ influx leads to tight-junction molecule degradation (VE-cadherin and occludin) and mitochondrial dysfunction in endothelial cells. (**C**) TRPM2-mediated Ca^2+^ influx is needed for immune cell migration and activation. During inflammation, ROS production in mitochondria is increased, which activates PARP in mitochondria or in the nucleus and enhances the production of ADPR. Increased ADPR potentiates TRPM2-mediated Ca^2+^ influx, which further increases the production of ROS in mitochondria, leading to the formation of a feed-forward vicious cycle. ROS-, PARP-, and Ca^2+^ -related signaling pathways increase the expression of proinflammatory genes, such as TNF-α, IL-1β, MCP1, and MIF. Moreover, TRPM2-mediated Ca^2+^ influx promotes cytoskeleton rearrangement and immune cell migration.

**Figure 4 cells-11-00491-f004:**
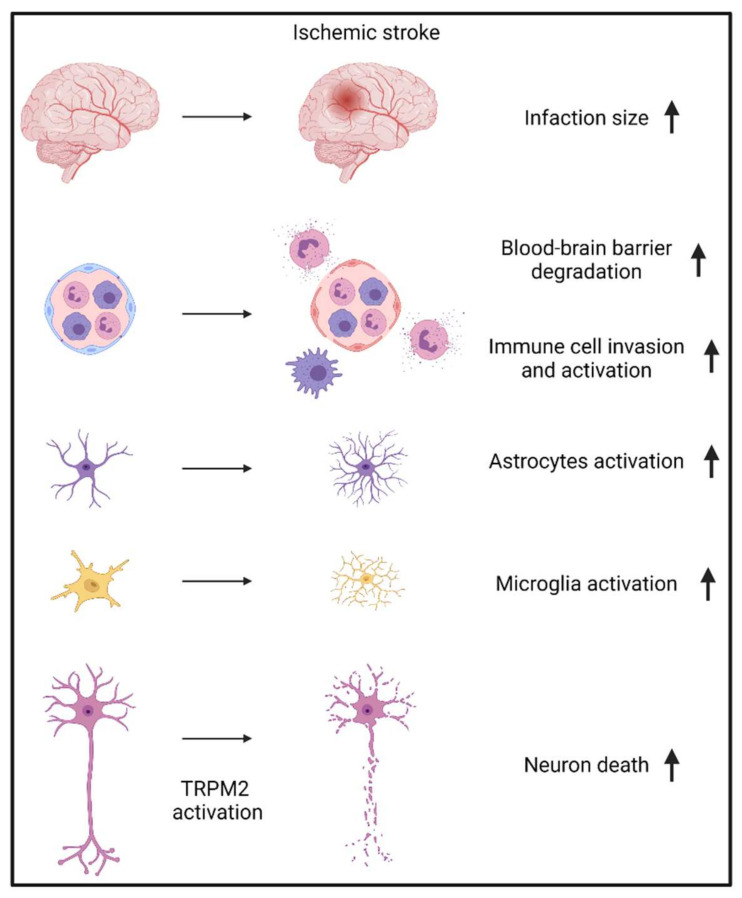
TRPM2 in ischemic stroke.

**Table 1 cells-11-00491-t001:** Summary of studies on the role of TRPM2 in aggregating ischemic brain/neuron injury.

Species	Model (s)	Target (s)	Mechanism (s)
Rat	In vitro H_2_O_2_ treatment	Neuron	Knockdown of TRPM2 using siRNA inhibited H_2_O_2−_induced neuronal death [181].
Mouse	In vitro OGD In vivo tMCAO	Neuron	Knockdown of TRPM2 using shRNA inhibited OGD-induced neuronal death, and reduced infarction size after MCAO [185].
Mouse	In vitro OGD In vivo BCCAO	Neuron	Global knockout of TRPM2 inhibited increase of intracellular Zn^2+^ and ROS production, and attenuated neuronal death after global ischemia [56].
Mouse	In vitro H_2_O_2_ treatment In vivo tMCAO	Neuron	Global knockout of TRPM2 inhibited increase of neuro-excitability in response to H_2_O_2_, and attenuated neuronal death and brain injury after tMCAO by promoting pro-survival signaling while inhibiting pro-apoptotic signaling [192].
Mouse	In vivo neonatal hypoxic ischemic brain injury model	Neuron	Global knockout of TRPM2 attenuated neuronal death and reduced infarct size after hypoxic–ischemic brain injury partially by regulating GSK-3β signaling [184].
Mouse	In vivo CA/CPR	Neuron	Inhibition of TRPM2 using clotrimazole reduced neuronal death in male mice, but not in female mice [193].
Mouse	In vivo tMCAO	Neuron	Inhibition of TRPM2 using a peptide inhibitor reduced infarction size after MCAO [194].
Mouse	In vivo tMCAO	Immune cells	Transplantation of bone marrow from global TRPM2 knockout mice into wild-type mice, or inhibition of TRPM2 using ACA reduced infarction size in wild-type mice after MCAO [55].
Mouse	In vivo BCAS	Microglia	Global knockout of TRPM2 inhibited brain damage and cognitive dysfunction [172].
Human	In vitro BSO treatment	Microglia and astrocytes	Knockdown of TRPM2 using siRNA attenuated the inflammatory responses in human microglia and astrocytes [171].

OGD, oxygen–glucose deprivation; BCCAO, bilateral common carotid artery occlusion; tMCAO, transient middle cerebral artery occlusion; CA/CPR, cardiac arrest and cardiopulmonary resuscitation; BCAS, bilateral common carotid artery stenosis; BSO, D,L-buthionine-S,R-sulfoximine.

## Data Availability

We choose to exclude this statement because this review did not report any data.

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
