# Peer review of "A Systemic Review of the Integral Role of TRPM2 in Ischemic Stroke: From Upstream Risk Factors to Ultimate Neuronal Death"

_cells, 2022, doi:10.3390/cells11030491_

Round 1

Reviewer 1 Report

This is a comprehensive review of the role for TRPM2 channel in the development of ischemic stroke and the progression of post-stroke neuronal dysfunction. There are several issues to be considered as follow:

  1. Elaborate the sentence one lines 72 to 74: definitions of ADPR1 and 2 as well as MHR1/2, hs, and dr.
  2. Order the figure numbers according their appearance.
  3. Lines 97 and 330: Trpm2 -- irregular usage.
  4. Line 172: elevation.
  5. Line 198: cholesterol included “in” oxidized…?
  6. Lines 201 and 203 state the same thing.
  7. Line 286: thrombi from deep vein cause pulmonary embolism but not ischemic stroke. Similarly, the sentence in line 321 should be revised: References 169 and 170 describe that the incidence of thromboembolic events increases in patients with leukemia but not that deep vein thrombosis results in ischemic stroke. Embolic ischemic stroke develops by the occlusion of cerebral arteries by thrombi formed in the heart or major arteries.
  8. Line 337: astrocytes are an essential component of NVU, please revise.
  9. Line 419: incomplete sentence.
  10. Line 469: incomplete sentence.
  11. Perspective is mostly the repetition of Introduction, and this part needs extensive revision.
  12. References: inconsistent styles.
  13. Figure legends should be typed but not embedded as graphics.

Author Response

This is a comprehensive review of the role for TRPM2 channel in the development of ischemic stroke and the progression of post-stroke neuronal dysfunction.

Response: Thanks for the positive comment!

There are several issues to be considered as follow:

1, Elaborate the sentence one lines 72 to 74: definitions of ADPR1 and 2 as well as MHR1/2, hs, and dr.

Response: We have added more details on this interesting finding, and added information on these abbreviations. MHR1/2 has been introduced at the beginning part of this section, which is a highly conserved domain among all the TRPM members.

2, Order the figure numbers according to their appearance.

Response: Our original order was right, which was disrupted by the automatic transformation during the submission. We have corrected the order.

3, Lines 97 and 330: Trpm2 -- irregular usage.

Response: Thanks for the careful proofreading! They were caused by the automatic transformation during the submission. We have corrected them.

  1. Line 172: elevation.

Response: Thanks for the careful proofreading! We have replaced it with the more suitable “increase”.

4, Line 198: cholesterol included “in” oxidized…?

Response: Thanks for the careful proofreading! Yes, we meant to use “included in”. We have corrected it.

5, Lines 201 and 203 state the same thing.

Response: We have corrected the redundant writing here.

6, Line 286: thrombi from deep vein cause pulmonary embolism but not ischemic stroke. Similarly, the sentence in line 321 should be revised: References 169 and 170 describe that the incidence of thromboembolic events increases in patients with leukemia but not that deep vein thrombosis results in ischemic stroke. Embolic ischemic stroke develops by the occlusion of cerebral arteries by thrombi formed in the heart or major arteries.

Response: We have removed the description regarding thrombi formed in deep vein at line 286 and line 321.

7, Line 337: astrocytes are an essential component of NVU, please revise.

Response: We have replaced the “glial cells” with more accurate “astrocytes”.

8, Line 419: incomplete sentence.

Response: Thanks for the careful proofreading! We have corrected it!

9, Line 469: incomplete sentence.

Response: Thanks for the careful proofreading! We have corrected it!

10, Perspective is mostly the repetition of Introduction, and this part needs extensive revision.

Response: We have extensively revised the perspective section to add more depth in discussion.

11, References: inconsistent styles.

Response: Thanks for the careful proofreading! This was caused by the automatic transformation during the submission. We have corrected it.

12, Figure legends should be typed but not embedded as graphics.

Response: We have remade the figures, and added the text figure legends in the manuscript.

Reviewer 2 Report

This is a good review on the role of TRPM2 in brain ischemia. I have only two concerns:

the subparagraph in cancer should be remouved  and another paragraph in the role of other channels of the same family in stroke should be added. 

Author Response

This is a good review on the role of TRPM2 in brain ischemia.

Response: Thanks for the positive comment!

I have only two concerns:

the subparagraph in cancer should be removed and another paragraph in the role of other channels of the same family in stroke should be added. 

Response: We have removed the subparagraph in cancer, and added another paragraph in the role of other TRPM channels in stroke.

Reviewer 3 Report

Dear Authors,

thank you for the comprehensive review of literature and your own work considering this interesting topic.

However I would like to suggest some corrections:

General remark:

please put some more effort to basic understanding of stroke patophysiology when making final statements, also it is to recommend to make final conclusions of a chapter with more probability and less certainity especially if this is based on own not published work.

  1. considering stroke ethiology - TOAST classification should be used
  2. Line 242 - I suggest to put this statement a bit less strong as this can not be proven without any doubts
  3. Line 286 - extracranial thrombus from deep vein is not a cause of stroke without PFO, so can not be put as one
  4. Line 309 - please document the statement that "artery thrombus formation due to cancer compression" can lead to stroke
  5. Line 328/329 - please give some proof that treating leukemia by targeting TRPM2 can reduce stroke incidence or put the statement in some more "probable" way
  6. Line 340 - document Figure 4 with studies
  7. Line 359 (statement and discussion) - please explain/documment the simmilarity between endothelial cells of the lung and endothelial cells which form blood-brain barrier 
  8. Table 1 - shows results of which studies?
  9. Line 469/470 - is first sentence finished?
  10. Please pay attention to some misspelling through the paper

Thank you. Wishing you a successfull future work.

Author Response

Dear Authors,

thank you for the comprehensive review of literature and your own work considering this interesting topic.

However, I would like to suggest some corrections:

General remark:

please put some more effort to basic understanding of stroke pathophysiology when making final statements, also it is to recommend making final conclusions of a chapter with more probability and less certainity especially if this is based on own not published work.

Response: Thanks for the excellent suggestions! We have made the claims from our own unpublished work less strong to avoid over-stretch statements.

1, considering stroke ethiology - TOAST classification should be used

The TOAST classification denotes five subtypes of ischemic stroke: 1) large-artery atheroscle- rosis, 2) cardioembolism, 3) small-vessel occlusion, 4) stroke of other determined etiology, and 5) stroke of undetermined etiology.

Response: Thanks for this valuable information! We have added the TOAST classification in the beginning paragraph of the 2nd section.

2, Line 242 - I suggest putting this statement a bit less strong as this cannot be proven without any doubts

Response: We have changed the strong word to “is likely to”. Our work regarding the role of TRPM2 in atherosclerosis has experienced a lengthy review process, and finally is almost accepted by nature cardiovascular research. It will be online in the March issue if everything goes well.

3, Line 286 - extracranial thrombus from deep vein is not a cause of stroke without PFO, so can not be put as one

Response: We have removed deep vein thrombus related contents in our manuscript.

4, Line 309 - please document the statement that "artery thrombus formation due to cancer compression" can lead to stroke

Response: We have removed the cancer section based on reviewer#2’s suggestion.

5, Line 328/329 - please give some proof that treating leukemia by targeting TRPM2 can reduce stroke incidence or put the statement in some more "probable" way

Response: We have removed the cancer section based on reviewer#2’s suggestion.

6, Line 340 - document Figure 4 with studies

Response: The related studies in Figure 4 is summarized in Table 1. We have added “as summarized in Table 1” where we mentioned Figure 4.

7, Line 359 (statement and discussion) - please explain/documment the simmilarity between endothelial cells of the lung and endothelial cells which form blood-brain barrier

Response: Endothelial cells comprising the neurovascular unit are very different to their counterparts in the peripheral, as we mentioned a bit earlier in this subsection, with the reference of this comprehensive review paper demonstrating the substantial differences between central and peripheral endothelial cells (Tight junctions of the blood-brain barrier, U Kniesel and H Wolburg, 2000).

8, Table 1 - shows results of which studies?

Response: Table 1 summarizes the studies related to the role of TRPM2 in aggregating neuron damage and/or brain injury using various cellular and animal models (as shown in the table legend).

9, Line 469/470 - is first sentence finished?

Response: Thanks for the careful proofreading! We have finished this sentence.

10, Please pay attention to some misspelling through the paper

Response: We have carefully proofread our manuscript, and corrected some misspelling previously not identified.

Thank you. Wishing you a successful future work.

Response: Thank you so much! Also wish you a happy and productive 2022!

Round 2

Reviewer 1 Report

There are few additional suggestions:

  1. Figure 1 should be quoted in the main text before quoting Figure 2.
  2. Superscript 2+ for Ca2+ in Figures 2 and 3.
  3. Line 495: “its high permeability to zinc” is somewhat strange, isn’t it “its high capacity (or affinity?) to zinc for permeation” (?)

Reviewer 3 Report

To my best knowledge, all remarks have been accepted and paper accordingly changed.

Thank you for the cooperation and congratulation for your work.